# High-Performance Cataluminescence Sensor Based on Nanosized V_2_O_5_ for 2-Butanone Detection

**DOI:** 10.3390/molecules25153552

**Published:** 2020-08-04

**Authors:** Run-Kun Zhang, Jing-Xin Wang, Hua Cao

**Affiliations:** 1Guangdong Provincial Engineering Research Center of Public Health Detection and Assessment, School of Public Health, Guangdong Pharmaceutical University, Guangzhou 510310, China; wjx@gdpu.edu.cn; 2School of Chemistry and Chemical Engineering, Guangdong Pharmaceutical University, Zhongshan 528458, China

**Keywords:** cataluminescence, V_2_O_5_, 2-butanone, gas sensor

## Abstract

The development of high-performance sensors is of great significance for the control of the volatile organic compounds (VOCs) pollution and their potential hazard. In this paper, high crystalline V_2_O_5_ nanoparticles were successfully synthesized by a simple hydrothermal method. The structure and morphology of the prepared nanoparticles were characterized by TEM and XRD, and the cataluminescence (CTL) sensing performance was also investigated. Experiments found that the as-prepared V_2_O_5_ not only shows sensitive CTL response and good selectivity to 2-butanone, but also exhibits rapid response and recovery speed. The limit of detection was found to be 0.2 mg/m^3^ (0.07 ppm) at a signal to noise ratio of 3. In addition, the linear range exceeds two orders of magnitude, which points to the promising application of the sensor in monitoring of 2-butanone over a wide concentration range. The mechanism of the sensor exhibiting selectivity to different gas molecules were probed by quantum chemistry calculation. Results showed that the highest partial charge distribution, lowest HOMO-LUMO energy gap and largest dipole moment of 2-butanone among the tested gases result in it having the most sensitive response amongst other VOCs.

## 1. Introduction

Volatile organic compounds (VOCs) in residential buildings and in the workplace pose serious threats to public health [1,2,3]. As a typical colorless VOC, 2-butanone is produced industrially on a large scale, is widely used as a common solvent, denaturing agent and cleaning agent. It finds wide use in the manufacture of plastics, textiles, paints and other coatings [4]. However, short-term inhalation exposure to 2-butanone causes irritation to the eyes, nose, and throat of humans, and even central nervous system depression [5]. Slight neurological, liver, kidney, and respiratory effects have also been observed in chronic inhalation studies of 2-butanone in animals. Taking measures to protect public health, the Occupational Safety and Health Administration has promulgated an 8-h permissible exposure limit (PEL) of 590 mg/m^3^ (200 ppm) for 2-butanone [6]. Therefore, the development of a simple and low-cost method for rapid monitoring of 2-butanone in air environments, especially in workplaces, is of significant importance to avoid its potential hazard.

Gas sensors represent a versatile technology for harmful gas monitoring because of their advantages of low-cost, small size and easy operation [7,8,9,10]. Nowadays, various gas sensors based on different principles, such as electrochemistry [11,12,13,14], surface acoustic wave [15,16], quartz crystal microbalance [17,18], and cataluminescence (CTL) [19,20,21], have been designed for the detection of different analytes. Among them, gas sensors based on CTL have attracted a great deal of scientific interest. CTL is a phenomenon of light emission produced by the catalytic oxidation of analyte on solid nanomaterial surface. During the process of CTL emission, only sample and oxygen in air are consumed; the solid nanomaterial is not consumed [22,23,24]. This characteristic means that CTL-based sensors have the advantage of good stability that is essential to long-term monitoring. With the development of material science, many advanced nanomaterials have been introduced into the design of CTL-based sensor. For instance, F-doped cage-like SiC-based sensors for hydrogen sulfide [25], Y-doped metal-organic framework-5-based sensors for iso-butanol [26], sensors based on hydrotalcite-supported gold nanoparticles for acetaldehyde [27], and sensors based on Y_2_O_3_ with multi-shelled hollow structure for methanol [28]. The expanded availability of nanomaterials has greatly advanced the development of CTL-based sensors.

Vanadium oxide (V_2_O_5_) has gained much research interest because of its excellent structural flexibility, lower bandgap and high energy density [29,30]. These features enable V_2_O_5_ to be utilized for numerous applications including gas sensing. Nanosized V_2_O_5_ with different structures have been synthesized for the electrochemical sensing of ethanol [31], ammonia [32] and xylene [33]. However, the application of nanosized V_2_O_5_ in the design of CTL-based sensors has seldom been reported. In this paper, nanosized V_2_O_5_ was synthesized by a simple hydrothermal method. The as-prepared V_2_O_5_ was found to be a promising candidate for CTL sensing of 2-butanone. The CTL sensor based on V_2_O_5_ exhibited high sensitivity, good selectivity, rapid response speed and recovery speed to 2-butanone. Other VOCs, including acetone, 3-pentanone, *n*-hexane, methanol, ethanol and so on, only produced a very weak or no response when they flowed through V_2_O_5_ surface. It is well known that CTL-based sensors have a good selectivity due to the harsh conditions for CTL emission. However, the gas-sensing selectivity mechanism remains uncertain. Herein, the possible gas-sensing selectivity mechanisms were investigated based on quantum chemistry calculation.

## 2. Results and Discussion

### 2.1. Characterizations of Nanosized V_2_O_5_

Figure 1 shows the XRD patterns of nanosized V_2_O_5_ and precursor before calcination. The XRD pattern of precursor shows a poor crystallinity, the diffraction peaks cannot be assigned to any known single compound, meaning the precursor may be an adduct, or a complex mixture of oxides and hydroxides. For the XRD pattern of nanosized V_2_O_5_, all the diffraction peaks match the orthorhombic crystalline phase of V_2_O_5_ (JCPDS card no. 41–1426).

The grain sizes of V_2_O_5_ is was calculated by using the Debye-Scherrer equation [34]:(1)$$D=\frac{\kappa \lambda }{\beta \cos2\theta }$$
where κ is the shape factor and its value is 0.94, λ is the wavelength of X-ray radiation (1.5406 Å), β is the full width at half the maximum intensity and θ is the Bragg angle in Radian. The *d*-spacing (Å) was calculated by using Bragg’ equation [35]:(2)$$d=\frac{n\lambda }{\beta \sin2\theta }$$
where *n* is the order of diffraction equal to 1. The dislocation density (nm^−2^) was calculated according to the following equation [35]:(3)$$\delta =\frac{1}{D^{2}}$$

The microstrain (ε) was calculated by using the following equation [36]:(4)$$\varepsilon =\frac{\beta }{4\tan\theta }$$

Multiple peak fit was performed to determine the θ and β values from the XRD data. The results are summarized in Table 1. The average grain size was calculated to be 30.46 nm. The average *d*-spacing, δ and ε were determined to be 2.330 Å, 9.832 × 10^−3^ nm^−2^ and 5.275 × 10^−3^, respectively.

The lattice constants were determined according to the following formula [37]:(5)$$\frac{1}{d^{2}}=\frac{h^{2}}{a^{2}}+\frac{k^{2}}{b^{2}}+\frac{l^{2}}{c^{2}}$$
where h, k, l are the miller indices. Crystal planes of (200), (002) and (001) were chosen to determine lattice constants, and a, b and c were calculated to be 1.155 nm, 0.356 nm and 0.4373 nm, respectively. There calculated values are in very good approximation to standard values. [29,31]

Lower magnification the transmission electron microscopy (TEM) image (Figure 2a) shows the nanosized V_2_O_5_ displays a nearly hexagonal structure. Figure 2b narrates the high-resolution TEM (HRTEM) image of nanosized V_2_O_5_. The *d*-spacing of 0.44 nm is attributed to the (001) plane of V_2_O_5_. The selected area electron diffraction (SAED) shows the ring pattern mainly arising from the (001) and (002) planes of V_2_O_5_ structures, indicating the crystalline nature of V_2_O_5_ nanoparticles. Energy-dispersive X-ray spectroscopy (EDAX) with color mapping analysis was carried out to gain insight into the chemical composition and positional distribution of V and O in V_2_O_5_, the results are shown in Figure 2d–f. The results of the XRD pattern and TEM images indicate that highly crystalline V_2_O_5_ has been successfully synthesized.

### 2.2. Optimization of Sensing Conditions

CTL is produced during the heterogeneous catalytic reaction emitting photoemissions, the CTL spectrum usually is a continuum emission band ranging from the ultraviolet to blue wavelength. Under the conditions of operating temperature at 231 °C and flow rate of 220 mL/min, the influence of detection wavelength on sensing of 2-butanone was investigated in the range of 400–535 nm. Figure 3a shows the dynamic response curves of 2-butanone measured by using different interference filters. The maximum apparent CTL intensity (*I*) is observed at 460 nm of wavelength. The CTL response signal (*S*), background noise (*N*) and signal to noise ratio (*SNR*) at different wavelengths are plotted in Figure 3b. It can be seen that although the maximum *S* value appears at 460 nm, the background noise emitted by thermal radiation increases with the increase in wavelength, resulting in *SNR* reaching its maximum at 440 nm. Therefore, 440 nm was selected as the optimal wavelength for the CTL sensing of 2-butanone.

The operating temperature plays a critical role in gas sensing. Figure 3c shows the dynamic response curves of 2-butanone measured at different operating temperatures. It can be seen that the *I* value increases with the increase in operating temperature before 237 °C, and then decreases when the operating temperature is above 237 °C. The Figure 3d shows the change trends of *S*, *N* and *SNR* versus operating temperatures. It shows that the *S* value increases almost exponentially with increasing operating temperature until 237 °C, and then decreases significantly with a further increase in operating temperature. According to the classical theory, the luminous intensity is proportional to luminous efficiency and reaction rate [38]. Therefore, the change trend of *S* value perhaps can be attributed to the reaction rate, increasing exponentially with increasing operating temperature, but higher operating temperatures accelerate the molecular motion that leads to quenching of CTL emission. In addition, the background noise emitted by thermal radiation also increases with the increase in operating temperature, rendering a decrease in *S* value and *SNR*. Because the maximum *SNR* is observed at 231 °C, a temperature of 231 °C was selected as the optimal operating temperature for the CTL sensing of 2-butanone.

The influence of flow rate on the CTL sensing of 2-butanone was investigated by changing the flow rate ranging from 150 to 225 mL/min. All the 2-butanone samples were introduced into the sensor 4 s after data acquisition. The dynamic response curves of 2-butanone at different flow rates are shown in Figure 4a. It can be seen that the peak width of the dynamic response curve decreases as the flow rate is raised. The response time (*t_Res_*) and recovery time (*t_Rec_*) are two important performances to a gas sensor. Here, the response time is defined as the time taken to reach the maximum response signal after injection of sample, and the recovery time is defined as the time taken for maximum response signal to recover to baseline.

In order to further evaluate the relationship between flow rate and sensing performances, the dynamic response curve at each flow rate was measured in three replicates, the change trends of *t_Res_* and *t_Rec_* versus flow rate are shown Figure 4b, and the relationship between *S* value and flow rate is shown in Figure 4c. Both *t_Res_* and *t_Rec_* decreases as the increasing of flow rate, and then almost remain stable. For *S* value, it increases almost proportionally with increasing flow rate before 220 mL/min, and then decreases with a further increase in flow rate. These results show that the catalytic oxidation reaction of 2-butanone on nanosized V_2_O_5_ surface is controlled by diffusion rate when the flow rate is below 220 mL/min; that is the total reaction rate is controlled by the rate of 2-butanone in the gas phase transfer to the catalyst surface, and thereby the increase in flow rate causes increasing *S* value. However, the total reaction rate is controlled by the oxidation rate of 2-butanone on the nanosized V_2_O_5_ surface when the flow rate exceeds 220 mL/min, resulting in *t_Rec_* remaining stable. In addition, under high flow rate conditions, 2-butanone molecules flow through the catalyst surface too quickly to be sufficiently oxidized, rending a decrease in *S* value. In the case of 200 mg/m^3^ of 2-butanone, the short response time of 2 s and quick recovery time of 9 s were observed under 220 mL/min. Therefore, 220 mL/min was used for the sensing of 2-butanone as the sensitive response, as well as the satisfactory response and recovery time under this condition.

### 2.3. Evaluation of Selectivity

Selectivity is a very crucial performance for a gas sensor. The selectivity of a gas sensor can be defined as [39]:(6)$$\mathrm{Selectivity}=\frac{S_{\mathrm{other}\;\mathrm{analyte}}}{S_{\mathrm{target}\;\mathrm{analyte}}}\times 100\%$$

The selectivity of nanosized V_2_O_5_ sensors was investigated by measuring 200 mg/m^3^ of 2-butanone and 19 other kinds of gases commonly existing in workplace. Results showed that only 2-butanone, acetone, 3-pentanone and *n*-hexane produce CTL responses. The dynamic response curves of 2-butanone, acetone, 3-pentanone and *n*-hexane on nanosized V_2_O_5_ surface are shown in Figure 5a. A much stronger response of 2-butanone was observed. Figure 5b displays the selectivity of the V_2_O_5_-based sensor calculated by equation 6. The CTL responses of acetone, 3-pentanone, *n*-hexane are about 8.8%, 1.4% and 0.8% of that of 2-butanone, respectively. Other gases, including *n*-heptane, methanol, ethanol, ethyl acetate, benzene, toluene, *o*-xylene, *m*-xylene, *p*-xylene, formaldehyde, acetaldehyde, trichloromethane, tetrachloromethane, propyl acetate and ammonia, cannot induce a response under this condition. The above results fully demonstrate that the CTL sensor based on nanosized V_2_O_5_ has a good selectivity when sensing 2-butanone.

### 2.4. Analytical Characteristics

Figure 6a displays the CTL response signal of V_2_O_5_ toward different concentrations of 2-butanone under. A good linear relationship between the response signal and the concentration of 2-butanone was observed in the range of 0.5–600 mg/m^3^ (0.17–203 ppm). The linear regression equation is *S* = 117.4*c* + 177.7 (the correlation coefficient *r* = 0.9961), where *S* is the CTL response signal, and *c* is the concentration of 2-butanone. The limit of detection (LOD) at an *SNR* of 3 is 0.2 mg/m^3^ (0.07 ppm). The Occupational Safety and Health Administration permissible exposure limit of 2-butanone during an 8-h workday is 590 mg/m^3^ (200 ppm) [6]. The LOD of the present sensor is much lower than the permissible exposure limit; moreover, the linear range exceeds two orders of magnitude, indicating that the sensor has a potential for routine monitoring of 2-butanone in the workplace where 2-butanone levels may vary in concentration from low to the occasional high level.

The relative standard was 4.2% for five times sensing of 2-butanone at 200 mg/m^3^ (Figure 6b), indicating that the sensor has a good reproducibility. The analytical characteristics of some sensors for 2-butanone are summarized in Table 2. Compared with the other sensors for 2-butanone, the present sensor has a wider linear range and a lower LOD.

### 2.5. Gas-Sensing Selectivity Mechanism

The gas sensing process usually relates to the surface adsorption and surface reaction of gas molecules on the metal oxide nanoparticle. When metal oxide nanoparticles are heated in air, O_2_ molecules from the air can be adsorbed on the surface of metal oxide nanoparticle, and then chemisorbed oxygen species (O^2−^, O^−^, and O^2−^) are formed by trapping electrons from the conduction band [4,42]. The working temperature has an important impact on the type of chemisorbed oxygen species. An increase in temperature, the chemisorbed oxygen adsorbed on the surface of metal oxide nanoparticle undergoes the following state:$$O_{2}\left(\mathrm{gas}\right)\to O_{2}\left(\mathrm{ads}\right)$$
$$O_{2}\left(ads\right)+e^{-}\to O_{2}^{-}\left(ads\right)$$
$$O_{2}^{-}\left(ads\right)+e^{-}\to 2O^{-}\left(ads\right)$$
$$O^{-}\left(ads\right)+e^{-}\to O^{2-}\left(ads\right)$$

Previous studies demonstrated that O^−^ is the most active and stable chemisorbed oxygen species in the range of 100–300 °C [38]. Upon exposure to VOCs, VOCs can be oxidized into CO_2_ by the chemisorbed oxygen species on the surface of metal oxide nanoparticle [43]. If an excited intermediate was formed during the above process, and the excited intermediate returned to the ground state via radiative transition, CTL emission could be observed. From Figure 2, it can be seen that nanosized V_2_O_5_ exhibits a sensitive CTL response to 2-butanone, and a weak response to acetone, 3-pentanone and *n*-hexane. Although it is very difficult to identify the luminous species due to the very short-lived excited states, we can conclude that the catalytic reaction of the 2-butanone is more favorable to produce CTL emission than other VOCs (e.g., acetone, 3-pentanone and *n*-hexane) on nanosized V_2_O_5_ surface.

The sensing response is affected by the reactivity and absorptivity of gas molecules on sensing materials. In order to probe the mechanism behind the sensor shows different response to different VOCs, quantum calculations were implemented, and 2-butanone, acetone, and 3-pentanone and *n*-hexane were selected as representatives. All the optimization of structures and analysis of vibrational frequencies (energy calculation) in this study were performed using Gaussian 09 with B3LYP/6-311G (d, p) basis set.

The 3D molecular structures of the four molecules and their partial charge distribution calculated by Gaussian are shown in Figure 7. It can be seen that C1 site in 2-butanon, C1/C3 sites in acetone, O1 site in 3-pentanone, and C1/C6 sites in *n*-hexane are the more active sites to occur reactions due to these sites has higher charge dense than other sites in corresponding molecular structures. Because C1 site in 2-butanon has highest charge dense (−0.3249 e), meaning that 2-butanone is easier to react with the chemisorbed oxygen species than other gas molecules.

It has been revealed that the highest occupied and lowest unoccupied molecular orbitals (HOMO and LUMO) play a prominent role in governing chemical reaction. A large energy gap (*E*_g_ = *E*_HOMO_-*E*_LUMO_) between HOMO and LUMO implies high stability, which indicates low chemical reactivity. In turn, a small energy gap implies low stability and thereby high chemical reactivity [44]. Figure 8 illustrates the energy gaps in graph forms respective to the HOMO and LUMO levels of four molecules. The HOMO-LUMO energy gaps for 2-butanone, acetone, and 3-pentanone and *n*-hexane are 6.11, 6.18, 6.39 and 9.71 eV. It means that in order of reactivity from highest to lowest is 2-butanone, acetone, and 3-pentanone and *n*-hexane. A high level of consistency exists between the energy gap and the CTL response signal. Moreover, it was reported that the absorptivity of gas molecules on sensing materials is associated with the dipole moment of molecule in its gaseous state. The larger the dipole moment, the higher the attractive force between the gas molecule and sensing material [45,46]. The dipole moments in the molecular structure of 2-butanone, acetone, and 3-pentanone and *n*-hexane were calculated to be 2.7497, 2.6475, 2.6246 and 0 D, respectively. The dipole moment of 2-butanone is larger than other gas molecule, suggesting the interaction between 2-butanone molecules and nanosized V_2_O_5_ is strongest among these gases. Therefore, more 2-butanone molecules can react with the chemisorbed oxygen species on the nanosized V_2_O_5_ surface, displaying the observed selective response toward 2-butanone.

According to the above discussion, we can conclude that the catalytic oxidations of 2-butanone, acetone, and 3-pentanone and *n*-hexane on nanosized V_2_O_5_ surfaces are able to form excited intermediates and favor their radiative transitions. The highest partial charge distribution, lowest HOMO-LUMO energy gap and largest dipole moment of 2-butanone among the tested gases are responsible for its strongest sensing response. However, more research is still needed to better explain the detailed mechanism relating to excited state chemistry.

## 3. Experimental Section

### 3.1. Synthesis and Characterizations of Nanosized V_2_O_5_

Nanosized V_2_O_5_ was synthesized by a simple hydrothermal method. briefly, ammonium metavanadate solution (30 mL, 30 mg/mL) was mixed with dilute hydrochloric acid solution (20 mL, 1.0 mol/L) under vigorous stirring at room temperature till appearing bright yellow clarified liquid. The resultant solution was transferred into a 100 mL Teflon container. The Teflon container was sealed tightly by a stainless-steel autoclave, and then the autoclave was subjected to hydrothermal reaction at 180 °C for 12 h in a vacuum oven. The resultant precipitates were centrifuged and washed with deionized water and ethanol five times. Afterwards, the collected precipitates were dried at 100 °C for 10 h in a vacuum oven. Finally, the dried powders were calcined at 500 °C in the air for 2 h to obtain nanosized V_2_O_5_.

The morphology of the as-prepared nanosized V_2_O_5_ was characterized by TEM (FEI Tecnai G2 F20) with an accelerating voltage of 200 kV. Elemental analyses were performed by EDAX (Bruker Nano GmbH). The phase structure was investigated by Powder X-ray diffraction (XRD, Rigaku, Ultima IV) with Cu Kα radiation (λ = 1.5406 Å).

### 3.2. Fabrication of the Sensor System

The schematic diagram of the sensor for 2-butanone is shown in Figure 9. Nanosized V_2_O_5_ was sintered onto a ceramic heater to form a catalyst layer. The ceramic heater was inserted into a home-made quartz tube (length = 10 cm, diameter = 0.6 cm) with gas-inlet and gas-outlet. The quartz tube was placed into a sample chamber with a nono-paque widow at the bottom. The temperature of catalyst layer was controlled by adjusting the output voltage of the temperature controller connecting to the ceramic heater, and the detailed temperature could be measured by thermocouple. A photomultiplier (PMT) was assembled into a cylindrical tube under the nono-paque widow for measurement of luminous signal. Interference filters were used to reduce the background noise, and the detection wavelength can be selected by inserting different interference filters between the quartz tube and PMT. The air and sample were delivered by an air pump. The sample reacted with the oxygen in air on the surface of nanosized V_2_O_5_ to produce light emission. A commercial BPCL ultra-weak luminescence analyzer was used record and process the signal intensity.

The peak intensity on dynamic response curves directly recorded by the instrument is defined as apparent CTL intensity. The CTL response signal (*S*) is equal to the apparent CTL intensity minus the background noise, and can be expressed as:(7)S=I−N
where I is the apparent CTL intensity on dynamic response curves, N stands for the background noise. The signal to noise ratio (SNR) is described as:(8)$$\mathrm{SNR}=\frac{S}{N}$$

### 3.3. Methods for Quantum Calculations

Quantum calculations were implemented to reveal the mechanism of the difference of gas-sensing selectivity mechanism. Three compounds, 2-butanone, acetone, and 3-pentanone and *n*-hexane were selected as representatives in this theoretical study. All the optimization of structures and analysis of vibrational frequencies (energy calculation) in this study were performed by using Gaussian 09 with B3LYP/6-311G (d, p) basis set. After the structural optimization, Mulliken charges were obtained. Based on these values, the dipole moment (μ) along X, Y, and Z axis (μ_x_, μ_y_, and μ_z_) can be calculated. The dipole moment is calculated on the basis of the following equation:(9)$$(\mu )=\sqrt{u_{x}^{2}+u_{y}^{2}+u_{z}^{2}}$$

The energy of the HOMO) and LUMO were calculated based on the population analysis using Self-Consistent Field density.

## 4. Conclusions

In conclusion, V_2_O_5_ nanomaterials were successfully synthesized by a simple and facile hydrothermal method. CTL-based gas sensor fabricated by nanosized V_2_O_5_ shows high sensitivity, good selectivity, rapid response and recovery speed to 2-butanone. The detection conditions, including detecting wavelength, operating temperature and flow rate on the sensing of 2-butanone, were investigated in detail. Quantum chemistry calculation was performed to probe the gas-sensing selectivity mechanism. The CTL sensor based on V_2_O_5_ shows high sensitivity and selectivity to 2-butanone which can be attributed to 2-butanone having a highest partial charge distribution, the lowest HOMO-LUMO energy gap and the largest dipole moment. All these factors favor 2-butanone molecules reacting more easily with the chemisorbed oxygen species on the nanosized V_2_O_5_ surface. This work provides a promising method for the monitoring of 2-butanone in the indoor air environment, including in the workplace, in industry, and in houses.

## Figures and Tables

**Figure 1 molecules-25-03552-f001:**
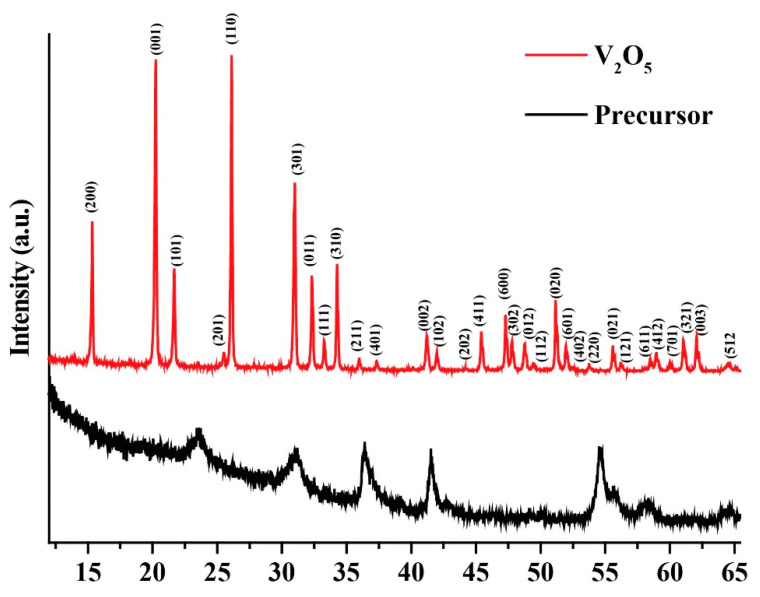
The XRD patterns of nanosized V_2_O_5_ and precursor before calcination.

**Figure 2 molecules-25-03552-f002:**
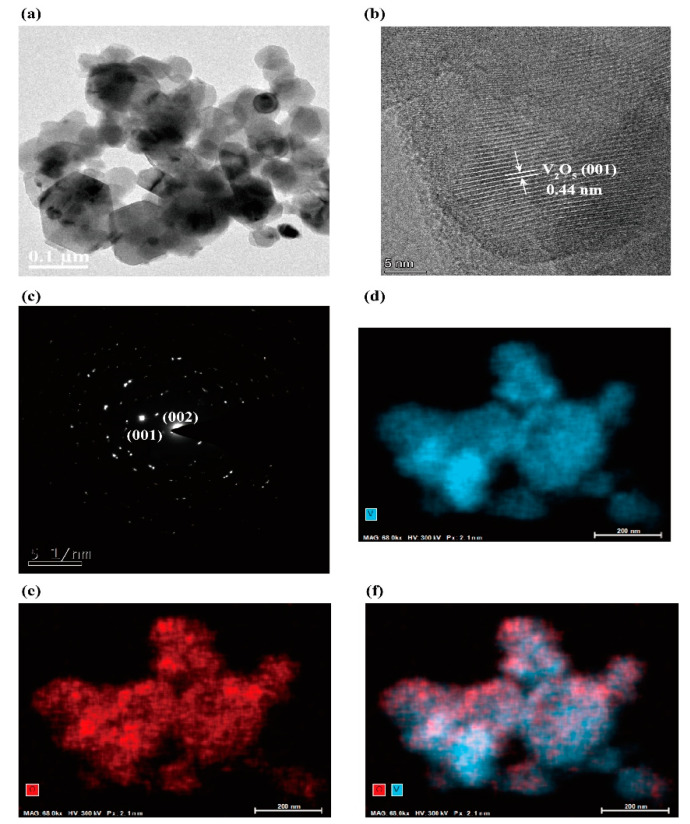
(**a**) TEM image, (**b**) HRTEM image, (**c**) SAED and EDAX color mapping of (**d**–**f**) of V_2_O_5_.

**Figure 3 molecules-25-03552-f003:**
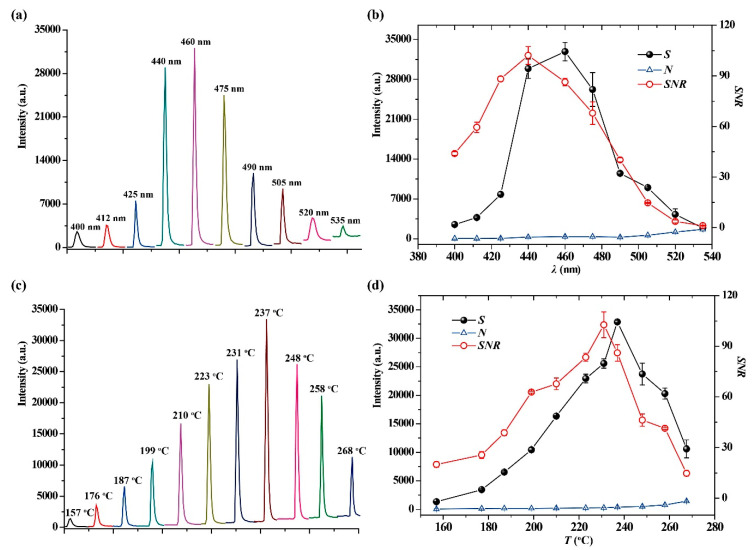
(**a**) The dynamic response curves of 2-butanone at different detection wavelengths. (**b**) The change trends of the values of *S*, *N* and *SNR* versus detection wavelengths. (**c**) The dynamic response curves of 2-butanone at different temperatures. (**d**) The change trends of the values of *S*, *N* and *SNR* versus temperature. Concentration of 2-butanone is 200 mg/m^3^.

**Figure 4 molecules-25-03552-f004:**
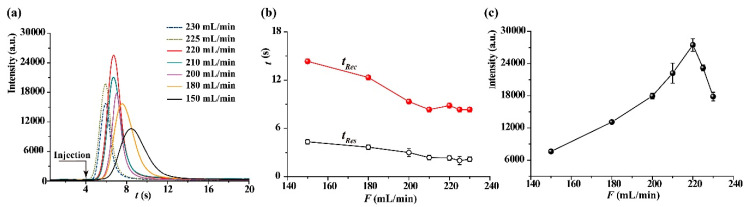
(**a**) The dynamic response curves of 2-butanone under different flow rates. (**b**) The change trends of *t_Res_* and *t_Rec_* versus flow rate. (**c**) The relationship between *S* value and flow rate. Wavelength: 440nm, temperature: 231 °C, the concentration of 2-butanone is 200 mg/m^3^.

**Figure 5 molecules-25-03552-f005:**
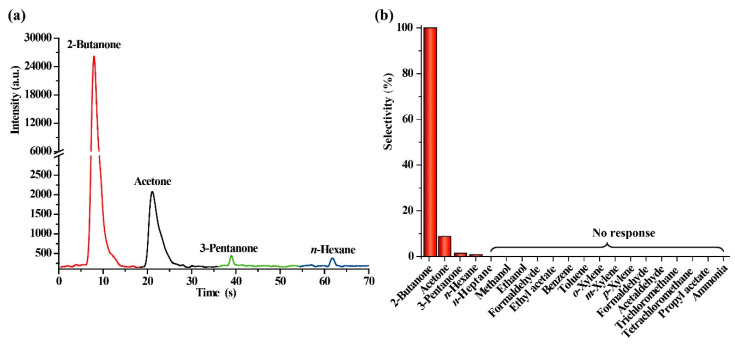
(**a**) The dynamic response curves of 2-butanone, acetone, 3-pentanone and *n*-hexane on nanosized V_2_O_5_ surface. (**b**) The selectivity of 2-butanone sensor based on nanosized V_2_O_5_. Wavelength: 440nm, temperature: 231 °C, flow rate: 220 mL/min, the concentration of all gases is 200 mg/m^3^.

**Figure 6 molecules-25-03552-f006:**
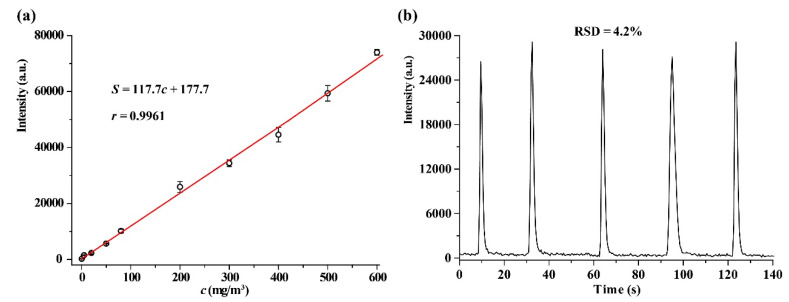
(**a**) Linear relationship between the CTL response signal and the concentration of 2-butanone. (**b**) Results of six replicate determinations of 2-butanone. Wavelength: 440nm, temperature: 231 °C, flow rate: 220 mL/min.

**Figure 7 molecules-25-03552-f007:**
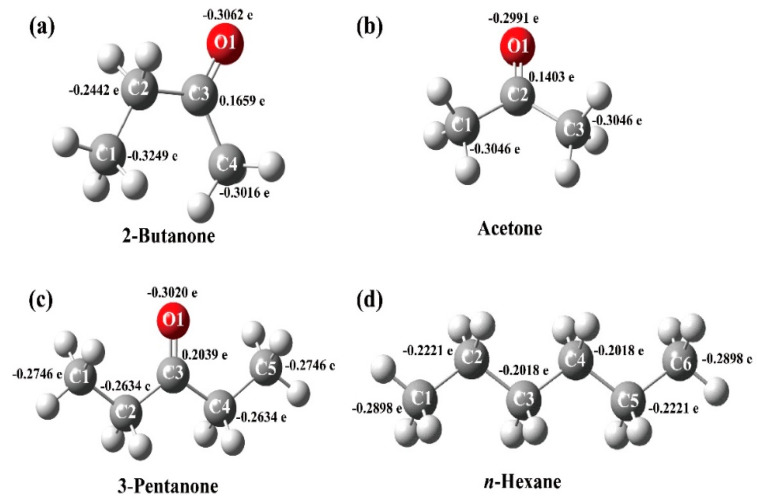
The 3D molecular structures and the partial charge distribution in molecules of 2-butanone (**a**), acetone (**b**), and 3-pentanone (**c**) and *n*-hexane (**d**).

**Figure 8 molecules-25-03552-f008:**
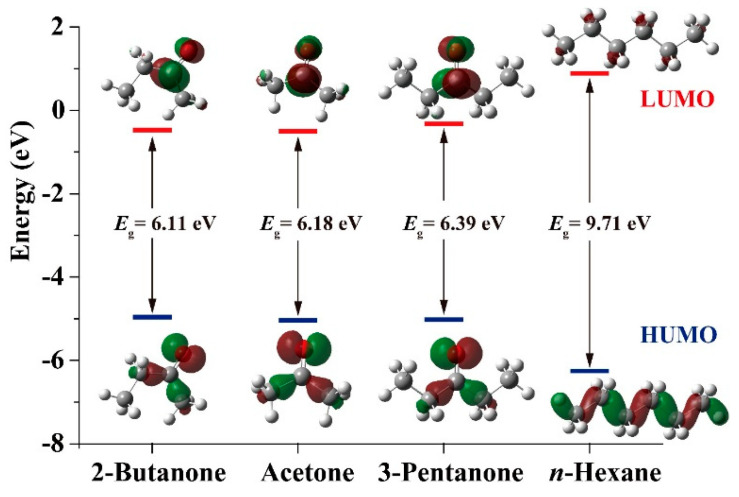
HOMO/LUMO energy levels, HOMO–LUMO gaps and orbital coefficients for 2-butanone, acetone, and 3-pentanone and *n*-hexane from B3LYP/6-311G (d, p) calculations.

**Figure 9 molecules-25-03552-f009:**
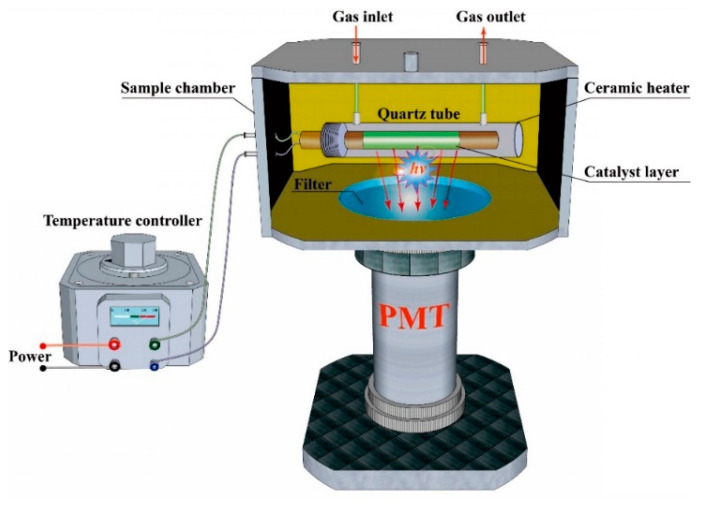
Schematic diagram of CTL-based sensor.

**Table 1 molecules-25-03552-t001:** The grain sizes, *d*-spacing, dislocation density and micro strain of nanosized V_2_O_5_. calculated from the XRD data.

No.	2θ (^o^)	Crystal Planes	β (^o^)	*D* (nm)	*d*-Spacing (Å)	δ × 10^−3^ (nm^−2^)	ε × 10^−3^
1	15.33	(200)	0.2812	28.51	5.777	1.231	9.119
2	20.29	(001)	0.3049	26.47	4.373	1.427	7.434
3	21.68	(101)	0.2726	29.67	4.096	1.136	6.212
4	25.43	(201)	0.6638	12.27	3.500	6.643	12.84
5	26.09	(110)	0.2572	31.71	3.412	0.994	4.842
6	30.99	(301)	0.2598	31.73	2.883	0.993	4.088
7	32.33	(011)	0.2517	32.86	2.767	0.926	3.789
8	33.30	(111)	0.2407	34.44	2.689	0.843	3.513
9	34.38	(310)	0.3570	23.29	2.606	1.843	5.035
10	36.01	(211)	0.1961	37.80	2.492	0.551	2.632
11	37.35	(401)	0.1351	83.92	2.406	0.260	1.744
12	40.13	(311)	0.6619	12.78	2.245	6.125	7.907
13	41.22	(002)	0.2392	35.49	2.189	0.794	2.775
14	41.94	(102)	0.1983	42.91	2.152	0.543	2.257
15	44.25	(202)	0.5745	14.93	2.045	4.488	6.166
16	45.39	(411)	0.2626	32.79	1.997	0.930	2.740
17	47.33	(600)	0.2868	32.51	1.919	1.093	2.855
18	47.84	(302)	0.2932	29.64	1.900	1.138	2.884
19	48.78	(012)	0.2805	39.56	1.865	1.034	2.699
20	49.51	(112)	0.2962	90.94	1.840	1.146	2.803
21	51.20	(020)	0.2663	33.08	1.783	0.914	2.425
22	52.05	(601)	0.2992	29.55	1.756	1.145	2.673
23	52.59	(402)	0.5593	15.84	1.739	3.984	4.939
24	53.74	(220)	1.4643	6.08	1.704	27.03	12.61
25	55.56	(021)	0.2473	36.31	1.653	0.758	2.048
26	56.22	(121)	0.2986	91.35	1.635	1.099	2.439
27	58.46	(611)	0.0980	95.82	1.577	0.116	0.764
28	59.00	(412)	0.2296	39.76	1.564	0.632	1.770
29	60.10	(701)	4.4027	2.08	1.538	230.1	33.21
30	61.19	(321)	0.3442	26.81	1.514	1.391	2.540
31	62.09	(003)	0.2417	38.36	1.494	0.680	1.752
32	64.58	(512)	1.0553	8.91	1.442	12.610	7.287
Average values				30.46	2.330	9.832	5.275

**Table 2 molecules-25-03552-t002:** Comparison of the gas sensing characteristics of sensors for 2-butanone based on different materials.

Principle	Materials	Temperature (°C)	Linear Rage (mg/m^3^)	LOD (mg/m^3^)	References
CTL	V_2_O_5_	231	0.5–600	0.2	Present work
CTL	Zn-doped SnO_2_	241	2310–92570	600	[40]
Electrochemistry	ZnO	400	5.9–295	1.2	[41]
Electrochemistry	WO_3_-Cr_2_O_3_	205	14.75–295	*ND*	[42]

*ND* stands for not discussion.

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
