# Peer review of "High-Performance Cataluminescence Sensor Based on Nanosized V2O5 for 2-Butanone Detection"

_molecules, 2020, doi:10.3390/molecules25153552_

Round 1

Reviewer 1 Report

Please see the attached file for detailed comments.

Reviewer 2 Report

In this paper, the authors have synthesized the V2O5 nanoparticles for a cataluminescence based gas sensor and showed a high sensitivity selectivity to 2-butanone gas. In addition, they tried to detailedly analyze V2O5 through various analysis methods such as an XRD and TEM and explained the sensing mechanism according to the energy gap between the HOMO and LOMO and dipole moment via quantum calculation. It's an interesting and effective report, but I have the following suggestions for improvements before the final acceptance of this paper.

  1. Line 71~76; The XRD results and calculated values are very well organized in Table 1, so it doesn't seem necessary to list every number in the text.
  2. Line 171; “In case of 20 mg/m3, ~~” It probably modified as 200 mg/m3. The authors should check the number.
  3. Line 249~250; I suggest to change the “organic compounds” as “volatile organic compounds (VOC)”.
  4. Line 249~250; The authors should briefly explain how the dipole moment for gas is calculated.
  5. I recommend that the Figures for the gas sensing data (figure 2~5) are replaced to be clearly visible.

Reviewer 3 Report

In this work, the authors presented a study about the cataluminescence sensing response of V2O5 nanomaterials. However, the manuscript shows several inconsistencies, which should be revised before the manuscript might be considered for publication, as follow:

The manuscript organization and writing require significant improvements. The selectivity results are shown before the sensing response optimization. This is not the proper way to discuss sensing results. Selectivity results are a consequence of the optimized sensing response, consequently, it should be presented after the sensing response optimization. The authors should make a rigorous revision of the manuscript writing and organization. 

The discussion of XRD results are very superficial and it is not be proper for a scientific manuscript (lines 70-76). In addition, Scherrer and d-spacing equations are in disuse to quantify the structure parameters of nanomaterials. The authors should improve the XRD discussion and present Rietveld refinement data, which are more adequate to analyze XRD patterns of nanomaterials.

The morphology of the nanomaterials was not well-characterized by the presented results. The authors should improve the morphology characterization including SEM and HRTEM data as well as a SAED pattern to corroborate the crystalline structure results.

The gas-sensing mechanism and its relationship with the improved butanone detection should be better discussed. Is the gas-sensing mechanism the only responsible for this enhanced sensing response to butanone? Any contribution from size and/or morphology parameters?

Is the V2O5 crystallized during the hydrothermal treatment or in the conventional calcination? If the V2O5 nanomaterials is crystallized during the hydrothermal treatment, why did the authors make a calcination treatment? If do not, did the authors characterize the material collected from the hydrothermal treatment? What was that?

Did authors compare the cataluminescence sensing response before and after calcination treatment? It is well-known on literature the influence of the calcination process on the sensing response of oxide nanomaterials.

Round 2

Reviewer 3 Report

The authors have improved the manuscript. It is suitable for publication.